**Perspective**

# Quantity and quality of minichromosome maintenance protein complexes couple replication licensing to genome integrity
Anoop Kumar Yadav [1,2] & Hana Polasek-Sedlackova [1] ✉

Accurate and complete replication of genetic information is a fundamental process of every cell division. The replication licensing is the first essential step that lays the foundation for error-free genome duplication. During licensing, minichromosome maintenance protein complexes, the molecular motors of DNA replication, are loaded to genomic sites called replication origins. The correct quantity and functioning of licensed origins are necessary to prevent genome instability associated with severe diseases, including cancer. Here, we delve into recent discoveries that shed light on the novel functions of licensed origins, the pathways necessary for their proper maintenance, and their implications for cancer therapies.

Error-free duplication of genetic information and its transmission across cellular generations is an essential prerequisite for the healthy life of any organism. Deoxyribonucleic acid (DNA), a carrier of genetic information, is replicated by the individual multiprotein molecular machinery, termed replisome, assembled at specific regions of chromosomal DNA referred to as replication origins. The length and complexity of genomes determine the quantity and chromosome distribution of replication origins, which vary among organisms. While the bacterial genome contains a single circular chromosome with one origin, the eukaryotic cells initiate DNA replication from thousands of origins scattered across the genome[1]. To avoid the generation of disease-causing alterations of genetic information, the DNA replication program defined by the number, distribution, and activity of replication origins must be tightly regulated to ensure that DNA is precisely replicated once during the S phase of the cell cycle. Any imbalance in the origin regulation can induce DNA damage and genome instability often associated with severe diseases, including cancer. The optimal quantity of replication origins and their proper distribution along the genome is directly regulated by a replication licensing system, which is the first step of eukaryotic DNA replication necessary for accurate and complete genome duplication[2].

During replication licensing, which occurs from late mitosis until the late G1 phase, replication origins are licensed-to-replicate by loading minichromosome maintenance (MCM) protein complexes onto DNA (Fig. 1a). The MCM complexes composed of six subunits—MCM2, MCM3, MCM4, MCM5, MCM6, and MCM7 (MCM2–7)—form the ring-shaped structural core of replicative helicase, the heart of all replisomes, which using ATP hydrolysis unwind DNA template to allow its

synthesis[3]. To ensure proper origin recognition and helicase loading, the origin licensing is tightly controlled by three major factors: origin recognition 1–6 (ORC1–6)[4–6], cell division cycle 6 (CDC6)[7–9], and CDC10-dependent transcript 1 (CDT1)[10–12]. ORC1–6 in complex with CDC6 first recognizes and binds the origin of DNA replication. In the case of budding yeast, replication origin is identified by a specific DNA sequence, while in higher eukaryotes, the origin recognition is rather driven by chromatin and epigenetic features. Once ORC-CDC6 has bound to the origin, it serves as a platform to recruit MCM2–7 in open ring conformation chaperoned by CDT1, resulting in the formation of the pre-replicative complex (pre-RC). The subsequent coordinated action of licensing factors leads to the stable encirclement of the DNA by MCM double hexamers, which are kept in the inactive state throughout the licensing period[13–16]. Upon entry to the S phase, inactive MCM2–7 complexes are activated by a process known as origin firing—a series of events including phosphorylation of MCM complexes by DBF4-dependent and cyclin-dependent kinases (DDK and CDK) and the attraction of various firing factors TRESLIN, MTBP, TOPBP1, RECQL4 (Sld3, Sld7, Dbp11, Sld2 in yeast)[17–24] and most recently identified DONSON as a specific firing factor in higher eukaryotes[25–30]. The concerted action of phosphorylation activities and firing factors leads to the recruitment of two binding partners, namely cell division cycle 45 (CDC45) and go-ichi-nii-san (5-1-2-3 in Japanese, GINS), each of which stably associates with one MCM2–7 ring resulting in the formation of the active replicative CMG (CDC45-MCM2–7-GINS) helicase[31–33]. Once CMGs are assembled, DNA at the replication origin is melted, and additional replisome components are recruited to establish two

[1]Department of Cell Biology and Epigenetics, Institute of Biophysics of the Czech Academy of Sciences, Brno, Czech Republic. [2]Department of Experimental Biology, Faculty of Science, Masaryk University, Brno, Czech Republic. ✉e-mail: polasek-sedlackova@ibp.cz

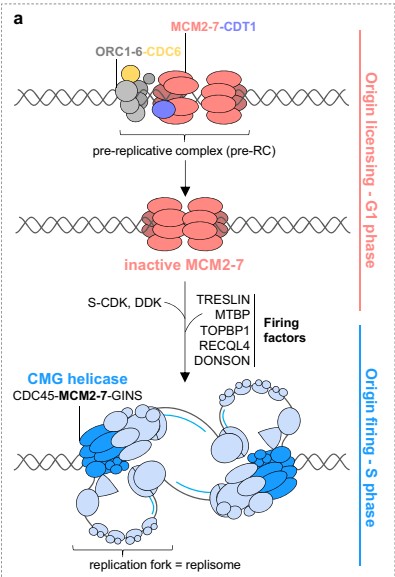

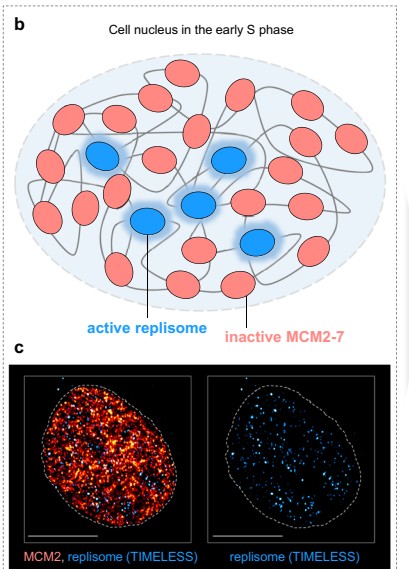

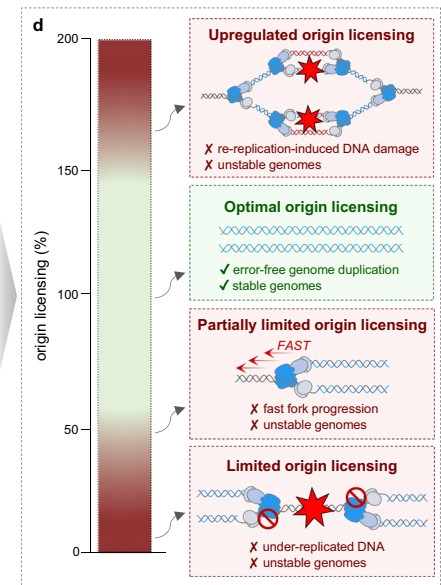

**Fig. 1 | Correct quantity and functioning of licensed origins are essential to maintain genome stability. a** Origin licensing is orchestrated by the licensing factors (ORC1–6, CDC6, and CDT1), which loads inactive MCM2–7 complex on chromatin in the G1 phase. In the S phase, inactive MCM complexes are converted to active replicative CMG helicases by the coordinated action of S phase-specific CDK, DDK, and firing factors. CMG helicases form the molecular motor of the two replisomes that move apart and duplicate DNA. **b** A graphical depiction of the cell nucleus in the early S phase depicting that only a minor fraction of loaded MCM complexes become activated during the genome duplication. **c** Representative confocal images of an early S phase nucleus of U2OS cells immunostained with MCM2 and TIMELESS antibodies. TIMELESS was used as a proxy to detect the replisome. DNA was stained with 4,6-diamidino-2-phenylindole. Scale bar 14 μm. **d** Proper and timely regulated origin licensing is essential to prevent genome instability. An upregulated level of licensing factors in the S phase can lead to re-replication, inducing DNA damage and genome instability. Partially limited origin licensing (when only an inactive MCM pool is affected) leads to the fast progression of replication forks that generate DNA damage and genome instability. Limited origin licensing can generate longer distances between incoming replication forks, eventually causing under-replicated DNA and genome instability in mitosis and subsequent cellular generations.

bidirectional replication forks moving from each other. The latest advances uncovering the molecular and structural basis of eukaryotic replication initiation are summarized in recent review articles[34–36].

Without a sufficient amount of MCM complexes loaded on DNA, genome duplication cannot be completed in a timely and accurate manner, but equally important is to shut down the licensing system before entry into the S phase (Fig. 1a–d). Re-licensing of replicated origins triggers the re-replication, leading to DNA breaks and other consequent gross chromosomal rearrangements. Tight cell-cycle-dependent regulation of ORC, CDC6, and CDT1 (reviewed in refs. [37,38]) ensures that MCM complexes can be loaded only during the licensing permissive cell cycle phase characterized by low CDK activity. Activation can then occur only upon high CDK levels in the S phase[39]. This cell cycle control of replication licensing guarantees that none of the origins can be licensed and activated more than once per cell cycle. On the other hand, the only chance to load helicase during the cell cycle puts pressure on the licensing system to succeed on the first attempt, which can be manifested by the loading of a higher level of inactive MCMs. Although the one-chance model is plausible, it raises several fundamental questions: How is the MCM surplus generated and maintained across multiple cell generations? Why is a massive excess of inactive MCMs loaded on DNA if only a small fraction of ~10–20% is needed for DNA replication? Are the inactive MCMs that are not converted to CMGs just wasted? Do they have any function inside or potentially outside of the DNA replication program?

Here, in this Perspective, we discuss that the huge MCM surplus is by no means a trivial waste of cellular resources but a result of a carefully orchestrated series of molecular events safeguarding genome integrity at multiple levels. We highlight that the novel functions of extra-licensed origins not only reveal exciting insights into MCM biology but also uncover fundamental regulatory pathways of genome duplication and cell division. Finally, we discuss how the deregulation of replication licensing in human cells contributes to oncogenic transformation and how it can be exploited in cancer therapies.

## The MCM life cycle

MCM proteins belong to the most abundant replication proteins produced inside a cell (Table 1)[40]. The E2F transcription factors regulate the MCM transcription, culminating in the G1/S transition of the cell cycle[41–43]. Subsequently, MCM protein production continues throughout the S phase, reaching the maximum levels in the later stages (Fig. 2a, b)[44,45]. These findings imply that high MCM levels are generated already in mother cells before entry to mitosis, allowing the rapid initiation of origin licensing in the late stages of mitosis, as observed by many studies[44,46,47]. During origin licensing, MCMs are progressively loaded on DNA, resulting in maximum licensing in the late G1 phase, with almost no MCM complexes remaining in the soluble nuclear fraction[47,48]. However, it should be noted that the maximum origin licensing during the G1 phase may vary between cell types and developmental stages. Once the S phase is initiated, the MCMs are gradually removed from the chromatin until they reach zero levels in the G2 phase (Fig. 2a, c)[47,49]. Incidentally, this remarkable cell cycle-specific dynamics of MCMs on chromatin allows a safe distinction between pre-replicative and post-replicative chromatin, as shown by many imaging studies[50,51]. Although the level of chromatin-bound MCMs fluctuates, the total protein pool remains relatively constant throughout the cell cycle (Fig. 2c, d)[47,52]. Even when MCMs are not bound to chromatin, they are exclusively localized in the nucleus except for the *S. cerevisiae* organism, in which MCMs are transported in and out of the nucleus in a CDK-dependent manner (Fig. 2a)[52–59]. As shown by many studies, the retention of MCM complexes in the nucleus largely depends on nuclear localization signals of MCM2 and MCM3 subunits[60,61]. Viewed from the perspective of a one-chance model for origin licensing, which must be tightly regulated to prevent under- or re-replication, one would assume that keeping MCM complexes in the cytoplasm must be a safer solution. But why do human cells take such a risk and spend the resources to maintain MCMs in the nucleus after DNA replication when they are not needed and will be ejected to the cytoplasm anyway once the nuclear envelope breaks down during mitosis? One possible explanation of this intriguing question could be that total MCM levels are

**Table 1 | Quantity of DNA replication proteins per HeLa cell**

| Protein name | Function | Copy number (per cell) | Concentration [nM] (per cell) | UniProt ID (Human) |
|---|---|---|---|---|
| MCM2 | Replicative helicase | 438538 | 364.11 | P49736 |
| MCM3 | Replicative helicase | 332775 | 276.29 | P25205 |
| MCM4 | Replicative helicase | 543760 | 451.47 | P33991 |
| MCM5 | Replicative helicase | 428777 | 356.00 | P33992 |
| MCM6 | Replicative helicase | 356210 | 295.75 | Q14566 |
| MCM7 | Replicative helicase | 745421 | 618.90 | P33993 |
| MCMBP | MCM binding protein | 463463 | 384.80 | Q9BTE3 |
| ORC1 | Licensing factor | 15669 | 13.01 | Q13415 |
| ORC2 | Licensing factor | 142167 | 118.04 | Q13416 |
| ORC3 | Licensing factor | 93647 | 77.75 | Q9UBD5 |
| ORC4 | Licensing factor | 46677 | 38.75 | O43929 |
| ORC5 | Licensing factor | 70383 | 58.44 | O43913 |
| ORC6 | Licensing factor | 13744 | 11.41 | Q9Y5N6 |
| CDC6 | Licensing factor | 5220 | 4.33 | Q99741 |
| CDT1 | Licensing factor | 3371 | 2.80 | Q9H211 |
| TRESLIN | Firing factor | 6505 | 5.40 | Q7Z2Z1 |
| MTBP | Firing factor | 9095 | 7.55 | Q96DY7 |
| TOPBP1 | Firing factor | 10211 | 8.48 | Q92547 |
| RECQL4 | Firing factor | 12017 | 9.98 | O94761 |
| DONSON | Firing factor | 3371 | 2.80 | Q9NYP3 |
| MCM10 | Firing factor | 11914 | 9.89 | Q7L590 |
| CDC45 | CMG helicase | 60019 | 49.83 | O75419 |
| GINS1 | CMG helicase | 18412 | 15.29 | Q14691 |
| GINS2 | CMG helicase | 133564 | 110.89 | Q9Y248 |
| GINS3 | CMG helicase | 72753 | 60.40 | Q9BRX5 |
| GINS4 | CMG helicase | 111379 | 92.47 | Q9BRT9 |
| TIMELESS | Replication progression complex | 32039 | 26.60 | Q9BVW5 |
| TIPIN | Replication progression complex | 35709 | 29.65 | Q9BVW5 |
| CLSPN | Replication progression complex | 13590 | 11.28 | Q9HAW4 |
| AND1 | Replication progression complex | 225707 | 187.40 | Q9HAW4 |
| POLE | Leading strand synthesis | 25611 | 21.26 | Q07864 |
| POLE2 | Leading strand synthesis | 10838 | 9.00 | P56282 |
| POLE3 | Leading strand synthesis | 51779 | 42.99 | Q9NRF9 |
| POLE4 | Leading strand synthesis | 28203 | 23.42 | Q9NR33 |
| POLA1 | Lagging strand synthesis | 56102 | 46.58 | P09884 |
| POLA2 | Lagging strand synthesis | 105356 | 87.47 | Q14181 |
| POLD1 | Lagging strand synthesis | 132851 | 110.30 | P28340 |
| POLD2 | Lagging strand synthesis | 397921 | 330.38 | P49005 |
| POLD3 | Lagging strand synthesis | 168165 | 139.62 | Q15054 |
| POLD4 | Lagging strand synthesis | 37904 | 31.47 | Q9HCU8 |
| PCNA | Core replisome component | 2595902 | 2155.30 | P12004 |
| RPA1 | Core replisome component | 926424 | 769.18 | O95602 |
| RPA2 | Core replisome component | 849920 | 705.66 | Q9H9Y6 |
| RPA3 | Core replisome component | 3507577 | 2912.23 | P35244 |

Data were extracted from the ref. [40]. The copy number of replication proteins may vary between cell types and developmental stages.

checked in the mother cells before the commitment to mitosis to ensure that newborn daughters instantly receive the same amount of MCMs. Alternatively, it may provide a certain level of plasticity under certain circumstances to prevent genome fragility[62] or promote gene amplification in specialized tissues[63].

A recent study uncovered an additional layer of complexity in the mechanism responsible for maintaining MCM levels in human cells[44]. Dual-HaloTag labeling protocol and time-lapse imaging approaches revealed that optimal MCM equilibrium is maintained by recycling and biogenesis pathways, which sustain optimal origin licensing but, importantly, give rise

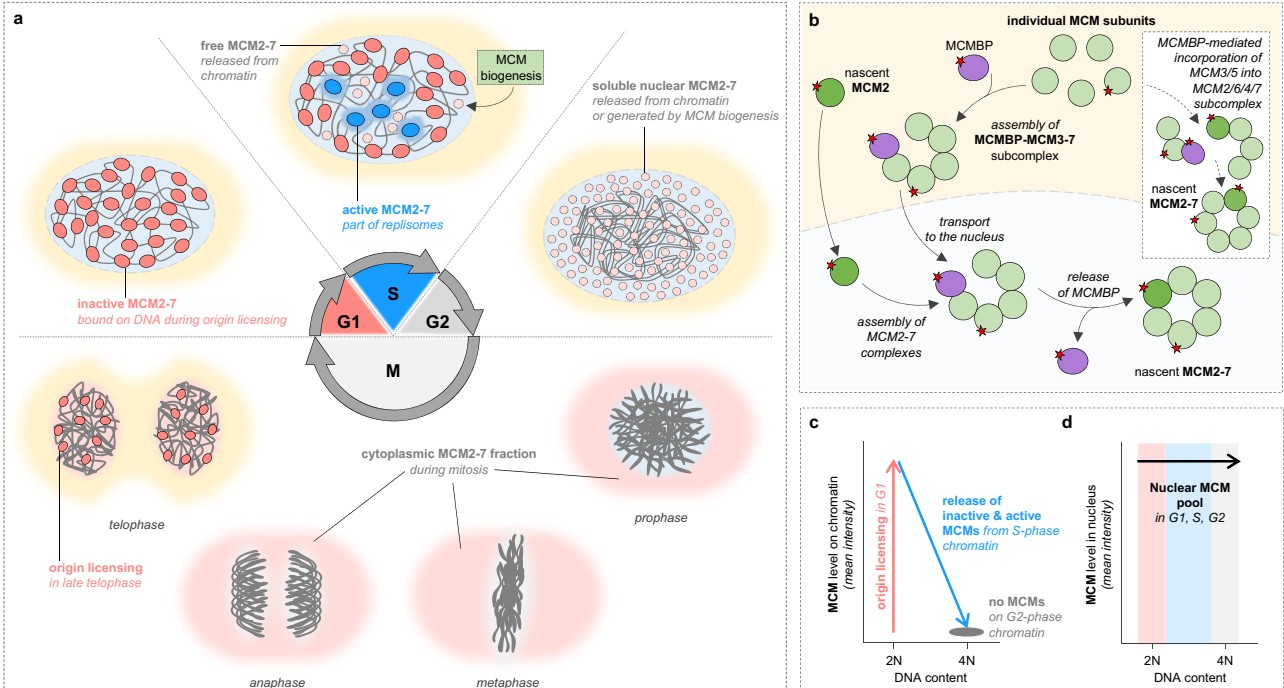

**Fig. 2 | The MCM life cycle. a** In the G1 phase, MCM2–7 complexes are rapidly loaded on chromatin by the replication licensing system. In the S phase, a minor fraction of loaded MCM2–7 is converted to active replisomes, which duplicate DNA, but moving through the chromatin, they also release unused inactive MCMs from chromatin. Throughout the S phase, the soluble pool of MCMs is increasing due to the replisome-dependent release of unused MCMs from chromatin as well as active CMG release during replication termination but also due to the active MCM biogenesis pathway. In the G2 phase, when the genome is fully duplicated, all MCMs are part of the soluble nuclear fraction. Once the nuclear envelope breaks down, MCMs are localized in the cytoplasm and are excluded from mitotic chromosomes during prophase, metaphase, and anaphase. In late mitosis, MCMs become localized in the forming nuclei, and the origin licensing of daughter cells is initiated. **b** Biogenesis of MCM complexes is initiated by the synthesis of individual MCM subunits by ribosomal apparatus in the cytoplasm. Currently, two models of MCMBP function

in this process have been proposed. The model proposed by Sedlackova et al.[44] suggests that MCMBP promotes the assembly of the MCM3–7 subcomplex and contributes to its rapid transportation to the cell nucleus. MCM2 enters the nucleus independently of MCMBP. Once all MCM subunits are in the nucleus, the formation of a complete MCM ring is initiated. During this, MCMBP is released from the MCM3–7 subcomplex, enabling the formation of complete MCM2–7 complexes that can eventually be loaded on DNA. The model proposed by Saito et al.[64] (inner box) suggests that MCMBP forms a complex with the MCM3/5 subcomplex and promotes its integration into a complete MCM ring. The red asterisk depicts nuclear localization signals on MCM subunits and MCMBP. **c** Flowcytometry-like profile summarizing MCM dynamics on chromatin during the individual phases of the cell cycle. **d** Flowcytometry-like profile summarizing MCM dynamics in the entire cell nucleus during the individual phases of the cell cycle.

to two different protein forms—parental and nascent MCMs (Fig. 3). Parental MCMs are defined as a protein pool involved in the DNA replication program in mother cells and subsequently reused in genome duplication of daughter cells. Nascent MCMs are newly synthesized in mother cells and first loaded onto chromatin in daughter cells (Box 1). Daughter cells inherit both MCM pools from their mothers in a ratio of approximately one parental to two nascent MCM rings. Although parental MCMs are less abundant in daughter cells, they are preferentially converted to active replicative CMG helicases, while the nascent MCMs remain primarily inactive. Beyond the functional differences of distinct MCM protein pools in the DNA replication program, which will be discussed in the following paragraphs, recent studies have also suggested differences in the molecular pathways sustaining the MCM equilibrium. One such finding involves the identification of MCM binding protein (MCMBP) as a specific chaperone for nascent MCM complexes (Fig. 2b)[44,64]. Specifically, it was shown that MCMBP strongly interacts with MCM sub-complexes and promotes its assembly into a complete hexameric ring[44,64–66]. Although the exact mechanism by which MCMBP assists in this process is not yet fully understood, two models have been proposed to explain the MCMBP function. The model proposed by Saito et al. suggests that MCMBP promotes the integration of the MCM3-MCM5 subcomplex into a complete MCM ring by directly interacting with nascent MCM3 and preventing it from proteasomal degradation[64]. While the proposed model is elegant in its simplicity, it appears to deviate from a previous study[66], in which authors identified a direct interaction between MCMBP and MCM7 subunit rather

than MCM3. This inconsistency could potentially imply that there may be another layer of intricacy in the role of MCMBP during MCM ring assembly. The model proposed by Sedlackova et al. suggests that MCMBP plays an important role in the assembly of nascent MCM3–7 subcomplex in the cytoplasm and facilitates its transport to the nucleus, while the MCM2 subunit enters the nucleus autonomously[44]. This model is supported by the observations that shortly after MCMBP depletion or mutating its nuclear localization signal (NLS), nascent MCM3–7 subunits are rapidly mislocalized in the cytoplasm while MCM2 remains orphaned in the nucleus[44]. These findings suggest that MCM3–7 and MCM2 are transported across the nuclear membrane in the form of two distinct subunits, similar to the transportation mechanism observed for ORC sub-complexes[67]. Whether separating the MCM2 subunit from the rest of the MCM subcomplex during its transport is a steric requirement or has a functional significance in the regulation of MCM ring assembly remains an interesting avenue for future investigations. Despite the gaps and discrepancies in current models of MCM biogenesis, both studies have concluded that MCMBP plays a pivotal role in the maintenance of MCM equilibrium necessary to carry out error-free genome duplication.

## Aligning MCM quantity with replication activity through rate-limiting factors

During the G1 phase, the replication licensing mechanism effectively deposits the majority of produced MCM complexes onto chromatin. Subsequently, a minor fraction of these complexes is converted into active

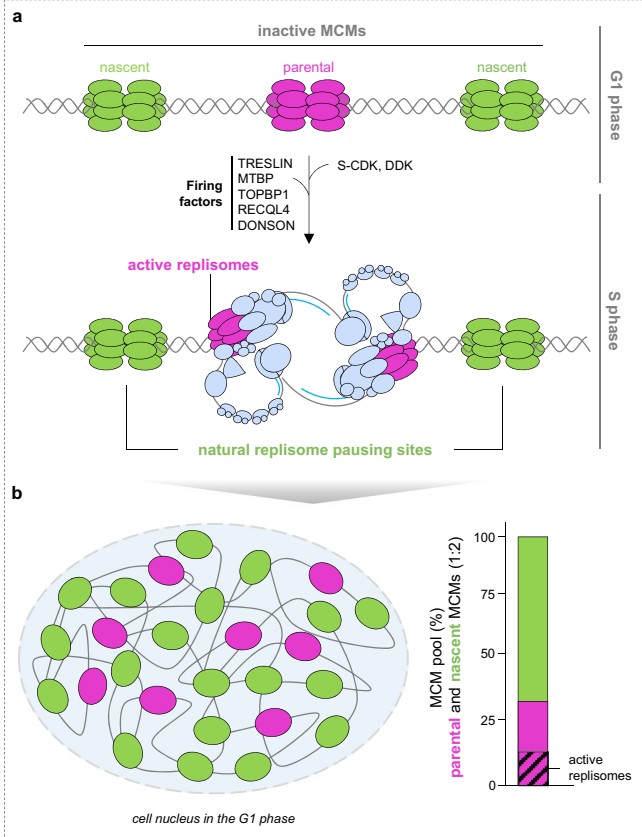

**Fig. 3 | The maintenance of MCM equilibrium by parental and nascent MCM helicases. a** Daughter cells inherit from their mothers both MCM pools. In the subsequent S phase, parental MCMs are preferentially converted to active replisomes, while nascent MCMs remain largely inactive but act as natural replisome pausing sites needed to adjust physiological replication fork progression. **b** During origin licensing in daughter cells, parental and nascent MCMs are loaded on chromatin in an ~1:2 ratio. The graph presents a general scheme that is based on findings in ref. 44.

---

## Box 1: | Glossary

**Origin licensing** is a process responsible for loading inactive MCM helicases on replication origins in a coordinated action of licensing factors such as ORC1–6, CDC6, and CDT1.

**Origin firing** is a process during which inactive MCMs are converted to active replication forks, basic units driving the genome duplication.

**Dormant (backup) origins** are origins that are repressed during normal DNA replication but can be activated by a checkpoint signaling pathway upon permanent stalling of two incoming replication forks.

**Parental MCMs** are defined as an MCM protein pool involved in the DNA replication program in mother cells and subsequently reused in genome duplication of daughter cells, where they are preferentially converted to active replicative helicases.

**Nascent MCMs** are defined as a newly synthesized MCM protein pool in mother cells and first loaded onto chromatin in daughter cells, where they remain primarily inactive but act as natural replisome pausing sites to set up physiological replication fork progression.

---

in replication domains[44,74]. Our calculations are consistent with numerous studies indicating that despite the variations between cell types and developmental stages, DNA-loaded MCM complexes greatly exceed the number of origins typically employed during the S phase[75–78].

Since its discovery, researchers have been perplexed by the purpose of excessive origin licensing during genome duplication (Fig. 1)[79,80]. The question of why cells invest valuable resources in loading MCMs through complex licensing mechanisms if they are not used has been challenging to answer, partly also because of the inability to observe MCM complexes at the sites of DNA synthesis inside a cell nucleus. These striking observations have become known as the MCM paradox. In Fig. 4, we outline the difficulties in detecting a small fraction of active MCMs and provide an explanation for this part of the MCM paradox. In the following sections of this Perspective, we delve into decades of research aimed at better understanding the importance and function of excessive MCM complexes during genome duplication.

### Checking the effectiveness of replication licensing

Although only a small portion of MCMs is converted into active replicative CMG helicases, the amount of loaded MCMs on DNA is constantly monitored by cell cycle control. If the licensed origins do not reach a certain threshold, G1 phase signaling will delay the entry into the S phase[81–87]. In primary cells, the delay of the G1 phase progression is characterized by low levels of CDK activity and hypo-phosphorylated RB (retinoblastoma). This can be achieved by different mechanisms. While one mechanism inhibits the transcription of the *cyclin D1*, the others target the CDK activity either through the induction of CDK inhibitors, such as p21 and p27, or by p53-dependent loss of phosphorylation in the activation loop of CDK2[83–86]. Although a tight connection between origin licensing and cell cycle regulation has been described, the exact mechanism by which cells sense the level of licensed origins is still unknown. As discussed previously[88], one possibility could be that origin licensing may, for instance, happen near the *cyclin D1* promoter, and the presence of MCMs could positively modulate the chromatin and transcription of *cyclin D1* (for more details, see paragraph Replication origins as modulators of chromatin and gene expression). Alternatively, if the ORC complex fails to load a sufficient number of MCMs, it could impede the natural turnover of ORC subunits on chromatin, potentially leading to the activation of the CDK activity suppression mechanism.

Despite the gaps in the proposed signaling, it is noteworthy to mention that all the above-described mechanisms were shown to operate independently of canonical p53 phosphorylation by ATM (Ataxia telangiectasia

---

replisomes in the S phase. Several studies have shown that critical components of origin firing and replication fork naturally limit the activation of replication origins[68–70]. Here, based on recent proteomic analysis[40], we provide detailed insight into the abundance of individual replication factors and their impact on origin activity. The protein copy number for individual MCM subunits ranges from 330,000 to 750,000 per HeLa cell. In contrast, the number of individual replication fork components or firing factors is several orders of magnitude lower, e.g., ~60,000 for CDC45, ~25,000 for POLE1, ~10,000 for POLE2, ~30,000 for TIMELESS, and ~3300–12,000 for firing factors (for more details see Table 1). Based on this, it is possible to estimate the number of replication fork components within a single replication domain defined as a 400–800 kbp genomic unit with 2–8 replication forks running synchronously[71]. On average, each replication domain contains ~13.6–27.2 copies of MCMs, ~2.4–4.9 copies of CDC45, and ~1.2–2.4 copies of TIMELESS. Interestingly, the levels of critical replication factors are considerably lower than the estimated number of replication forks per replication domain. This highlights the importance of efficient recycling of the core replisome components and firing factors. Recent research has indeed shown that recycling these crucial components is essential for the successful completion of genome duplication in both human cells and budding yeast[72,73]. However, it should be noted that under normal conditions, only a small fraction, ~25–30%, of the entire CDC45 and TIMELESS protein pool is used. Previous studies have revealed that forcing origin firing by inhibiting ATR (ATM- and Rad3-related kinase) or CHK1 (Checkpoint kinase 1) leads to a 3–4-fold higher accumulation of CDC45 or TIMELESS

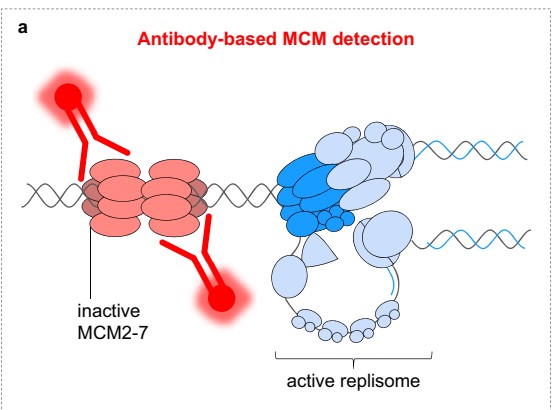

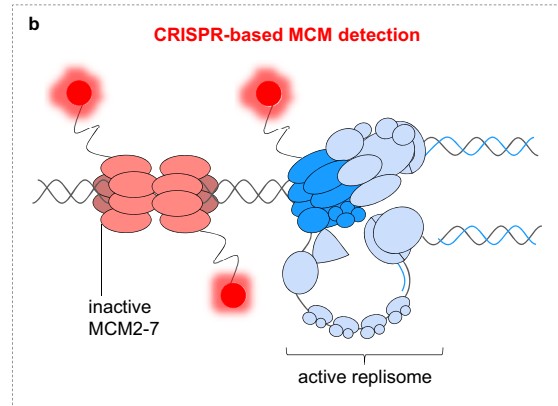

**Fig. 4 | Visualizing the heart of the replisome.** The complex biology of the MCMs has baffled researchers ever since their identification in the 1980s, through the discovery of their essential role in eukaryotic origin licensing in the mid-1990s, and up to the last decade marked by in vitro insights into MCMs as the structural core of the replicative helicase. The transfer of structural and biochemical knowledge of DNA replication to the in vivo settings has been hampered by the inability to detect MCMs at active replication sites in the cellular environment. In fact, most of the immunostaining data available in the literature reported rather the exclusion of MCMs from sites of active DNA replication[49,171–173]. Although few potential explanations considering alternative replisome architecture and limitations of visualizing techniques to detect a minor fraction of active MCMs have been proposed[49,172–176],

imaging of replisome dynamics in physiological settings has not been possible for several decades. In the era of CRISPR, a recent study provided an explanation of this paradox[74]. **a** Upon full assembly of the replisomes, the MCM scaffold is shielded by various interacting partners, limiting the access of anti-MCM antibodies, which primarily detect the inactive form of MCM complexes. **b** CRISPR-Cas9 endogenous tagging of MCM subunits by flexible linkers fused with fluorescent tags enables to overcome the sterical hindrance and allows visualization of both inactive MCMs as well as a minor fraction of active MCMs that are part of the replisomes. Although simple, this work opens new avenues to validate and study replisome architecture and its dynamics in the cellular environment using super-resolution microscopy or nanoscopy techniques.

mutated) or ATR kinases activated by the DNA damage response pathway[82–86]. Based on these observations, G1 signaling, in response to limited origin licensing, has been called the licensing checkpoint. Nevertheless, whether the licensing checkpoint is a unique signaling event or whether it is part of a complex G1 checkpoint network that, in addition to DNA integrity control, monitors the overall cellular readiness for proliferation needs to be investigated in the future. Additionally, it remains unclear how p53 becomes activated in response to limited origin licensing if there is no detectable DNA damage. Whether this is due to increased p53 stabilization or transient physiological stress signaling needs to be elucidated. However, depletion of p53 in primary cells can release the G1 arrest induced by the paucity of MCM loading on chromatin, underscoring the importance of p53 in licensing checkpoint[86]. Notably, previous studies have shown that numerous cancer cell lines with downregulated p53 are sensitive to the origin licensing level because of impaired licensing control. Entering the S phase with sub-optimal amounts of loaded MCMs results in DNA damage and death in cancer cells[82,86]. As such, targeting MCM equilibrium presents a promising approach to cancer therapy. Taken together, all aforementioned findings highlight the necessity of correct MCM quantity loaded on chromatin for appropriate cell proliferation and overall fitness.

## Plasticity in origin selection vs. firing capacity of individual origins
In the long term, the MCM surplus is vital for the fitness and survival of an organism[89–94]. Understanding the function of these extra-licensed MCMs has always been the subject of extensive research, and numerous models have been proposed. According to the stochastic model of origin activation[95,96], having a vast excess of the MCMs per single replication domain can increase the plasticity of origin selection, enabling DNA replication to adapt to various chromatin features or constraints. If an origin cannot be activated or is deleted, an adjacent origin may be activated instead, demonstrating how adaptable the selection process is[97,98]. Recent genome-wide sequencing has largely confirmed this stochastic activation, with no apparent sequence specificity in metazoans[99]. However, common epigenetic features associated with a subset of origins suggest that origin selection is not entirely random[96]. Chromatin features are an important criterion for origin selection; however, the biochemical properties of replication origins can also play a role in the selection process. Recently, it has been described that parental MCMs are preferentially converted to active replicative CMG

helicases despite being less abundant than their nascent counterparts (Fig. 3)[44]. Interestingly, this preference persists even when an ATR inhibitor is used to induce origin firing. The reason for this preferential activation is yet to be determined; however, it is plausible that parental MCMs are inherited with residual posttranslational modifications or are selectively deposited to highly efficient origins and thereby quickly attract firing factors and CMG components. Alternatively, it is important to note that not all licensed MCMs may have the same capacity to be activated, for instance, if the inactive MCMs adopt a structural conformation incompatible with origin activation. In line with this, recent in vitro study using purified *S. cerevisiae* proteins reported the presence of single MCM hexamers loaded on DNA[100]. However, whether such structures exist on in vivo chromatin remains to be determined. Although additional research is needed to gain insights into origin selection, the currently available evidence implies that the nature of chromatin-bound MCMs may be more intricate than the simplistic origin plasticity model.

## Activation of dormant origins in response to replication stress
Even though cells may survive and replicate their genomes with a limited amount of MCMs in the short term, numerous studies have reported that cells with limited origin licensing become hypersensitive to additional replication stress[101–106]. This implies that a fraction of inactive MCMs function as backup origins also referred to as dormant origins[107]. Typically, these origins are repressed during normal DNA replication, but they can be activated in response to replication stress that impedes replication fork progression. Upon complete inhibition of replication fork movement, ATR kinase and its downstream effector CHK1 become immediately activated and function as master regulators of the replication stress response[108]. They coordinate the stabilization and repair of stalled replication forks, as well as delay the cell cycle. Additionally, the ATR-CHK1 signaling pathway plays a pivotal role in regulating the dormant origin activation[106,109]. Many studies have shown that the inhibition of ATR or CHK1 triggers the unscheduled origin firing, leading to massive DNA damage and replication catastrophe[106,109,110]. If the rate-limiting factors of the DNA replication fork are exhausted, the integrity of replisome architecture becomes compromised, leading to DNA breakage and replication catastrophe-induced cell death. Although CHK1 effectively inhibits global origin firing, it still allows local origin activation within the currently active replication domain to

restore genome duplication[109]. It is currently unclear how CHK1 differentiates between active and later firing replication domains, but it is possible that the presence of specific substrates, such as firing factors, within the replication domain may facilitate the activation of dormant origins.

While discussing the mechanism of backup origin activation, it is necessary to note that under replication stress conditions, DNA replication forks have multiple mechanisms to prevent permanent fork stalling and the emergence of genome instability. During genome duplication, the DNA replication forks are constantly challenged by various extrinsic and intrinsic factors, leading to replication stress. Depending on the level of replication stress, cells employ adequate mechanisms to respond, for instance, adjusting the replication fork speed, employing DNA lesion bypass leading to daughter strand gaps repair behind the fork, or complex mechanisms of fork remodeling resulting in its restart. The molecular mechanisms underlying these replication fork responses are covered in recent excellent reviews[111,112]. In light of these recent findings, permanent replication fork stalling and checkpoint-mediated activation of backup origins can be viewed as one of the last attempts to rescue genome duplication of specific replication domains. Indeed, it has been suggested that backup origins may be activated only upon high levels of replication stress inducing permanent stalling of two incoming replication forks[107]. An intriguing example of the diverse mechanisms involved in the prevention and handling of under-replicated DNA arising from stalled replication forks are the common fragile sites (CFSs). CFSs are large chromosomal regions susceptible to recurrent breakage upon replication stress[113]. One notable observation regarding CFSs is that they inherently possess a low origin occupancy[114]. Several studies have shown that even if all attempts to restore DNA replication fail and some DNA remains under-replicated when the cell enters the mitosis, it is still not the end, as the cellular apparatus offers several chances to complete DNA synthesis, albeit outside of dedicated time[115–117]. As 'first chance' cells often employ mitotic DNA synthesis (MiDAS) to complete genome duplication[115,116]. However, if a considerable amount of under-replicated DNA escapes the MiDAS and is inherited by daughter cells, a 'second chance' mechanism delivered by 53BP1 nuclear bodies is activated[117]. The remarkable plasticity of the cellular apparatus seems to involve countless mechanisms that prevent or deal with the emergence of under-replicated DNA with the sole aim to safeguard genome stability. Another example of this could be a recent striking work demonstrating that the fragility of CFSs can be rescued by non-canonical S-phase licensing[62]. These findings, at the same time, challenge the widely accepted one-chance model of the replication licensing system. In the context of all of this, seeing the activation of backup origins as one of the plenty options to restore genome duplication immediately raises the question of whether the MCM excess loaded on chromatin exclusively functions as backup origins under replication stress or has any role under physiological conditions.

## Inactive MCM complexes as an integral component of replication fork speed control

The original concept of origin licensing is based on the premise that licensing must only occur once during the cell cycle and be successful[118]. Therefore, it is believed that the vast number of replication origins distributed across chromosomal DNA provides the needed flexibility in their activation to complete genome duplication within the allocated time. However, as discussed in several parts of this Perspective, only a small fraction of MCMs become activated as replicative helicases, while the majority of licensed origins never fire. Although the flexibility in origin activation is beneficial, the unused origins may also pose a threat to genome integrity because the MCM2–7 complex, even when inactive, it tightly encircles the DNA and thus can represent a physical barrier to incoming forks. Indeed, various biochemical studies using purified proteins from *S. cerevisiae* have shown that MCM double hexamers are extraordinarily stable structures resistant to very high salt concentrations washes[119–121]. Although in vitro experiments showed that inactive MCMs can be translocated on a non-chromatinized DNA template to distant sites by the CMG helicase[100,122], in vivo studies rather demonstrated frequent replication fork

pausing at the inactive replication origins in budding yeast[123,124]. As shown previously, physical obstacles interfering with replication fork progression belong to major sources of replication stress if left unresolved[125]. A wide range of sophisticated mechanisms has been described to resolve collisions in front of moving forks, including the mechanism evicting natural obstacles such as histones or pathways managing the resolution of DNA-protein-crosslinks[126,127]. However, how the incoming replication fork deals with inactive MCMs remains largely unexplored. While active CMG complexes are removed through CUL2(LRR1)-mediated ubiquitylation and p97-dependent pathway[128,129], removing inactive MCMs seems to require additional 5'−3' helicase activity complementing 3'−5' CMG unwinding activity. In budding yeast, Rrm3 and Pif1 helicases were shown to promote replication fork progression through sites with inactive origins[121,123]. However, it remains unclear whether a similar mechanism is needed to erase a massive excess of MCMs in human cells.

Unresolved physical impediments pose a serious threat to genome integrity, but on the other hand, natural replisome pausing sites scattered throughout the genome may be beneficial to achieve optimal replication fork speed and thereby mitigate genome instability. Recent studies described replication fork speed control as a first-in-line genome surveillance mechanism that protects replicating genomes against the amplification of physiological replication stress to more severe forms[130–132]. This mechanism enables rapid adjustment of the replication fork speed to a constantly changing cellular milieu, such as a fluctuating pool of deoxyribonucleotides (dNTPs) or a number of active replication origins[103,130–134]. Natural replisome pausing sites can represent another layer in replication fork speed control and can be viewed as 'DNA template mini checkpoints' controlling the duplication of individual genomic segments. From all-natural impediments, inactive MCM complexes seem to be the perfect choice for several reasons: (1) the number of loaded MCMs and their distribution along the chromosomal DNA; (2) coordinated removal by incoming replication fork; (3) the ability to pause ongoing fork; and (4) the potential to be activated under high levels of replication stress. Indeed, a recent study has shown that the partial removal of nascent MCMs, which remain primarily inactive during genome duplication, leads to pathologically accelerated replication forks and the generation of DNA damage (Fig. 1d)[44]. Slowing down the replication fork speed by removing the replisome accelerator TIMELESS or using mild aphidicolin (an inhibitor of DNA polymerases) resulted in the rescue of DNA damage, clearly indicating that fast forks are the source of this damage[44]. These findings support the notion that inactive MCMs act as natural replisome pausing sites and demonstrate that different mechanisms can be deployed to achieve physiological fork progression and thereby preserve genome integrity. In the absence of replication fork speed control, uncontrolled fast fork movement can potentially lead to detrimental issues, such as impaired fidelity of DNA polymerases, shortening the maturation time of Okazaki fragments, generating daughter strand gaps, disrupting the restoration of epigenetic information or sister chromatin cohesion, etc. Ultimately, the accumulation of such pathological replication intermediates increases the burden of DNA damage, leading to potentially devastating consequences for cell survival. Since cancer cells show severe addiction to replisome speed control mechanisms[130], this new knowledge about fork speed regulation could present new opportunities for cancer therapy.

## Replication origins as modulators of chromatin and gene expression

The considerable abundance of MCMs on chromatin during G1 and early S phases has always sparked interest regarding their potential broader functions within the cell. The observations that MCMs engage with diverse chromatin remodelers and transcription factors provide further support for the notion that they have the potential to impact both chromatin structure and gene expression. Early research indicated that MCMs are associated with chromatin more susceptible to nuclease digestion[135], consistent with a requirement for more relaxed chromatin at the replication origins. Accordingly, various histone remodelers, including HBO1, SNF2H, GRWD1, and PR-SET-7, have been reported to associate with replication

origins and contribute to origin licensing[136–140]. Although the mechanism of their action is not fully understood, they may be involved in chromatin organization promoting pre-RC formation or origin recognition itself. Interestingly, a recent study using budding yeast as a model organism revealed new evidence that ORC plays an important role in nucleosome organization at replication origins[141]. In addition to its MCM loader function, ORC instructs various chromatin remodelers to space nucleosomes regularly, which is crucial for efficient chromosome replication. Among the MCM subunits, the N-terminus of MCM2 is well-known for its tight association with histones and histone chaperones on chromatin, as part of the CMG complex, and in the soluble fraction[142,143]. Within the CMG complex, the MCM2 plays a critical role in a sophisticated mechanism that ensures symmetrical recycling of parental histones to sister chromatids[144]. Although MCM2's function as a histone chaperone at the replication fork is well understood, its interaction with soluble histones and chaperones in its free form, as well as its potential histone chaperone activity in inactive MCM complexes, remains largely mysterious.

The relationship between replication and transcription, the most essential cellular processes, has been extensively studied in recent years[145]. Numerous studies have demonstrated that the transcription machinery lacks the ability to disassemble inactive MCM double hexamers but is capable of pushing them away from their initial loading site[146–148]. These observations suggest that transcription may play an important role in the distribution of MCM along DNA. However, the relationship between transcription machinery and origin licensing may not be only one-sided. Early research suggested that MCMs may play a more direct role in transcription through association with RNA polymerase II or specific transcription factors. For instance, it has been shown that the MCM2 subunit interacts with the C-terminal domain of RNA polymerase II necessary for establishing protein-protein interactions throughout transcription and downstream processes[149]. Additionally, two studies have demonstrated that MCM5 interacts with the STAT1alpha transcription factor, which is required for the expression of interferon-gamma-responsive genes during the immune response[150,151]. Nevertheless, future studies will be needed to uncover the functional significance of these interactions and their impact on transcription machinery.

As MCM loading is a complex chromatin process during the G1 phase, it has the potential to directly or indirectly impact the organization of higher-order chromatin structures. The observation of direct interaction between cohesin and MCMs suggested their potential role in establishing higher-order chromatin[152,153]. A recent study by Dequeker et al. has demonstrated that MCM complexes can act as active barriers restricting cohesin-loop extrusion in the G1 phase[154]. However, a different model proposed by Emerson et al. suggests that cohesin can push licensed MCMs, leading to their localization at boundaries with a complex CTCF motif orientation[155]. This localization of MCMs at genetically determined boundaries may encode the position of highly efficient origins in the chromatin landscape of human cells. While the proposed models offer different perspectives on the role of MCMs in cohesion-loop extrusion, they represent exciting areas for further investigation.

## MCM quantity and functioning in cancer development and therapy

As discussed throughout this Perspective, precise and timely regulation of the replication licensing system is crucial for error-free genome duplication. Misregulation of origin licensing can result in replication-induced DNA damage, leading to genome instability often associated with severe diseases such as congenital diseases or cancer. Congenital diseases such as Meier-Gorlin syndrome or natural killer cell deficiency disease are associated with mutations in origin licensing or firing factors leading to dysfunctional DNA replication and accumulation of genome instability hallmarks. A recent review by Schmidt and Bielinsky provides a detailed description of the clinical phenotypes and functional mechanisms of these diseases[156]. In the last paragraphs of this Perspective, we focus on cancer as a common class of disease associated with genome instability and how the deregulation of

replication licensing contributes to its development. One of the initiating events in tumor development is the deregulation of oncogene activity. Cyclin E overexpression observed in a broad spectrum of human malignancies causes a reduction in the number of licensed origins, high levels of replication stress, and genome instability[157]. Indeed, insufficient MCM loading on chromatin could be one of the main causes of genome instability in cyclin E overexpressing cells. Mechanistically, such genome instability can be potentially fueled by the accumulated DNA damage induced by fast-fork progression and by the compromised ability to respond to additional replication stress[44,107]. Complementary to these findings, previous research has shown that a reduction in MCM levels can trigger cancer development in animal models[91–94]. Additionally, improper distribution of MCM on chromatin also contributes to genome instability in cyclin E overexpressing cells. As shown previously, a shortage of the G1 phase causes the activation of intragenic origins within highly transcribed genes, increasing replication-transcription conflicts associated with DNA double-stranded break formation and chromosomal re-arrangements[148]. Cyclin D1 is another oncogene that is frequently overexpressed in human cancers. This upregulation is typically caused by gene amplification, translocation, or increased protein stability[158]. Mutations in the C-terminal region of cyclin D1, associated with endometrial and esophageal cancer[159,160], cause nuclear retention of cyclin D1 and higher stabilization of CDT1 during the S phase[161]. Previous research has demonstrated that the upregulation of replication licensing machinery outside of the G1 phase can trigger DNA re-replication, leading to chromosomal rearrangements and genome instability[37]. Aligned with this, inappropriate expression of pre-RC components has been reported in the early stages of tumorigenesis in various cancers[162,163]. Although the exact mechanism behind this remains unclear, the upregulation of licensing factors may play a crucial role in tumor development and promote genome instability by origin re-licensing. In fact, overexpression of CDT1 or CDC6 can lead to tumor formation in mice[164,165]. All these findings highlight the importance of precise and timely regulation of replication licensing to prevent genome instability and cancer development.

While replication stress contributes to genome instability and tumorigenesis, it can also be harnessed for cancer therapy. Replication stress markers can serve as diagnostic tools, and small molecule inhibitors can target cellular pathways and factors to act as antitumor drugs[166]. Many chemotherapeutic drugs often target the process of DNA replication, including direct inhibition of dNTP supply, topoisomerases, DNA polymerases, or specific DNA repair pathways, which can increase toxicity and reduce cell survival. Numerous studies have shown that inappropriate expression of MCMs and other pre-RCs is a common feature of a variety of premalignant dysplasia and cancers[162,163]. Consistent with this notion, MCMs have been suggested as sensitive biomarkers in cancer screening and diagnosis[167]. In addition, targeting the MCM equilibrium with small-molecule inhibitors to either upregulate or downregulate it could be an effective anti-cancer approach (Fig. 1d). One possible strategy could involve the stabilization of licensing factors, such as CDT1, to trigger re-replication. Consistent with this, the downregulation of geminin, an inhibitor of CDT1, has been shown to cause re-replication-induced DNA damage and apoptosis in cancer cells[168]. Furthermore, it has been demonstrated that drugs like MLN4924, which stabilizes CDT1, can induce re-replication, resulting in cell death in checkpoint-deficient cancer cells[169]. This makes it a promising anti-cancer drug against p53 mutant tumors. The second potential strategy is to target molecular pathways that maintain the MCM equilibrium. Previous research has shown that many cancer cells are vulnerable to a reduction of licensed MCMs[82]. This, combined with observations that cancer cells utilize replication fork speed control to replicate their genomes at a slower pace[130], opens new possibilities in drug design. For instance, pharmacological inhibition of MCMBP or other hitherto unknown factors maintaining the MCM equilibrium can lead to a reduction of licensed origins and a higher burden of replication stress due to the fast fork progression. While

normal cells with active licensing checkpoint will temporarily arrest in the G1 phase, cancer cells will suffer from reduced origin licensing[82]. As suggested previously[2], the benefit of such inhibitors may be that origin licensing is essential for cell proliferation. Therefore, the adaptability of cancer cells to specific licensing inhibitors is less likely. All these considerations make replication licensing and its regulatory pathways an attractive target for anti-cancer therapy.

## Outlook

In 1984, Bik-Kwoon Tye's laboratory performed a genetic screen in budding yeast, from which MCM genes were identified as mutants defective in the maintenance of minichromosomes[170]. Forty years of active research have brought countless novel and unique insights into how MCM complexes, the heart of all replisomes, are assembled, activated, and function on DNA. All of this is thanks to the rapid development of powerful techniques such as in vitro reconstitution assays, cryo-electron microscopy, single-molecule approaches, CRISPR-Cas9 genome editing, next-generation sequencing techniques, etc. Many puzzles and paradoxes about MCM complexes have been solved. However, there are still some that remain unresolved or have emerged recently. For instance, How are replication origins recognized and regulated in higher eukaryotes? What are the most apical pathways needed for precise and timely origin licensing? How is the stoichiometry and assembly of MCM2–7 complexes regulated? Is the MCMBP the only MCM chaperone? How is the optimal level of replication licensing determined and communicated with cell cycle control? Is origin firing solely dependent on the stochastic model? Why are parental MCMs preferentially converted to active replisomes? Do all MCMs loaded on chromatin have the same capacity to be activated as replicative helicases? How are the unused inactive MCMs removed from the chromatin? Do MCMs possess any function outside of the DNA replication program? We believe that the combination of the above-mentioned state-of-the-art methodologies will help to gain more insights into these fascinating questions and underlying processes maintaining the MCM equilibrium necessary for error-free genome duplication. Given that MCMs are frequently overproduced in various forms of cancer, obtaining comprehensive molecular knowledge about the control of replication licensing may offer novel approaches in the field of cancer therapy.

## Reporting summary

Further information on research design is available in the Nature Portfolio Reporting Summary linked to this article.

## Data availability

Our paper contains only published literature. The data presented in Table 1 were extracted from ref. [40] based on mass spectrometry datasets deposited on the ProteomeXchange Consortium under accession numbers PXD002815 and PXD002829.

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

## Acknowledgements

This work was supported by grants from the Czech Science Foundation Junior Star (grant no. 22-20303M), the European Research Executive Agency under Horizon 2022 Widera Talent program (ERA grant agreement no. 101090292), and Jihomoravske centrum pro mezinarodni mobilitu (JCMM) project scholarship for foreign students. We thank J. Lukas and K. Somyajit for the critical reading of the manuscript. We thank all members of the Sedlackova lab for stimulating discussions and insightful comments on the manuscript.

## Author contributions

H.P.-S. devised the form and content of the manuscript, wrote the first draft, and prepared figures. A.K.Y. contributed to the writing of the manuscript and figure design and prepared the table. Both authors read and commented on the manuscript.

## Competing interests

The authors declare no competing interests.
