## [Peer Review File · Communications Biology]

Reviewers' comments:

Reviewer #1 (Remarks to the Author):

The MCM2-7 complex plays a central role in eukaryotic DNA replication. This complex binds genomic DNA during the late M to G1 phase, forming an MCM-double hexamer, the process known as licensing. It is subsequently converted into the CMG helicase for forming the replisome during the S phase. Multiple mechanisms control MCM2-7, ensuring that DNA replication takes place only once in the cell cycle to maintain genomic integrity. This review article by Yadav and Polasek-Sedlackova highlights the importance of regulating MCM2-7, with a particular focus on its high abundance compared to other replication factors. This review article is based on their previous reports (Ref. 40, Sedlackova et al. *Nature*, 2020; Ref. 75, Polasek-Sedlackova, *Nat. Commun.*, 2022), as well as numerous studies done by other researchers.

The review article is well-written and provides many recent updates on MCM biology. While there are no major issues, there are some points that the authors should consider for correction, clarification or additional information.

1. Yeast studies have driven the mechanistic study of DNA replication. Although the authors mainly discuss DNA replication and MCM biology in mammalian cells, there are instances where findings in yeast and mammalian cells are not clearly distinguished. Here are some examples.

- Line 49: "MCM double hexamers, which are kept in the inactive state throughout the licensing period (Ref. 10-12)". The references are from yeast studies. It should be noted that the human MCM-double hexamer was recently reported (Li et al. *Cell*, 2023).

- Lines 53-54: "most recently identified DONSON as a specific firing factor in vertebrates". Ref. 24 reported DONSON in *C. elegans*. It seems that DONSON is conserved even in insects and plants.

- Lines 59-60: "The recent development of in vitro reconstitution assays of eukaryotic DNA replication". As far as I understand, in vitro reconstitution has been performed with yeast proteins, not yet with human proteins.

2. The authors describe that MCMBP chaperones the MCM hexamer formation. However, there are discrepancies with other studies regarding the assembly of MCM2-7. The authors should address these discrepancies and provide a balanced view of the findings.

"One such finding involves the identification of MCM binding protein (MCMBP) that strongly interacts with the MCM3-7 subcomplex and acts as a chaperone for nascent MCMs (Ref. 40, 60-62). Specifically, MCMBP plays an important role in the assembly and nuclear transportation of nascent MCM3-7 subcomplexes, while the MCM2 subunit enters the nucleus autonomously (lines 126-130)".

"Additionally, it remains unclear why the MCM2 subunit is separated from the rest of the MCM subcomplex, which is chaperoned by MCMBP and transported to the nucleus (lines 134-136)".

Additionally, Figure 2b gives the impression that the assembly of MCM2-7 occurs only in the nucleus. I understand that these are based on the findings in their publication (Ref. 40. Sedlackova et al. *Nature*, 2020). However, there is no direct biochemical evidence showing that there are an MCM3-7 subcomplex and MCM2 monomer in normal conditions in this paper. There is another report showing there are subcomplexes of MCM2/4/6/7 and MCM3/5, which eventually form MCM2-7 (Ref.65, Prokhorova and Blow, *JBC*, 2000). In line with this study, a recent report showed that MCMBP directly binds MCM3, makes an MCM3/MCMBP subcomplex, and works to incorporate MCM3/5 into MCM2/4/6/7 (Ref. 60, Saito et al. *eLife*, 2022). This paper also argued that the nuclear transport function of MCMBP is not essential, and the assembly might occur both in the cytoplasm and nucleus.

3. In the section on MCM excess as an integral component of replication fork speed control, the authors give the impression that fork speed is mainly controlled by obstacles ahead of the forks.

However, it is well known that fork speed is strongly affected by the number of active origins and dNTP pools (Anglana et al. Cell, 2003; Poli et al. EMBO J. 2012; Zhong et al. JCB, 2013; Rodriguez-Acebes et al. JCB, 2018). The authors should acknowledge this point.

4. In the last section mentioning cancer development and therapy, there is a famous genetic disorder called Meier-Gorlin syndrome (MGORS) that the causative genes are found in many replication proteins, including MCM5 and 6. This point is relevant and should be included.

Reviewer #2 (Remarks to the Author):

In this Perspective, Yadav & Polasek-Sedlackova discuss how adequate origin licensing is a crucial step to ensure faithful DNA replication in S phase. They summarize the fundamental studies that established in the field mechanisms of origin licensing and firing, as well as some recent studies on MCM equilibrium and factors affecting origin selection in G1, and origin firing in S phase. They also discuss in detail dormant origins, which is an important safeguard mechanism against genome instability. The figures and table are attractive and helpful. This Perspective is very suitable for publishing in Communications Biology, and will be of interest to readers who are particularly interested in understanding more about origin licensing and firing, DNA replication, and genome stability. The authors are to be commended on a thorough presentation!

We noted several places where word-choice or phrasing could lead to confusion. The authors can consider making changes for clarity:

1. The protein concentrations in the Table are very useful for the field to have collected. However, line 151 refers to that paper as using single cell proteomics; the technology is not yet available to analyze whole proteomes of single (human) cells.
2. MCM complexes are referred to as helicases throughout the article – even in G1 phase -, although they are not yet helicases until they become part of the active CMG enzyme in S phase.
3. Line 142 – please clarify what is meant by the “soluble fraction”- as written it implies interactions away from DNA altogether, but the authors may have meant histones in their (bound) transit from parental to daughter strands.
4. The percentage of MCM that is loaded is influenced by MCM expression, and some tumor-derived cell lines express enough MCM that a substantial fraction remains unloaded. The authors may consider the consequences of different origin licensing protein expression levels in different cell types for this concept that most of the MCM is loaded. It may be worth mentioning that the HeLa cells used for the study supporting Table 1 have deregulated expression of most replication proteins.
5. The term “MCM excess” is used frequently, but the authors mean an excess of loaded MCM, and not necessarily just more MCM expression; they could avoid that potential confusion with minor wording changes.
6. In panel 2C, the figure labeling indicates that the reduction in total MCM chromatin loading is from the removal of excess MCMs. During S phase the active MCMs are also removed as forks terminate, so this label could be misleading.

Finally, The authors are encouraged to clarify how some citations directly match facts of the sentence in which they occur. For example, Line 333 refers to studies demonstrating fast fork progression, but ref 131 doesn't measure Okazaki fragments, only PARP dependence which is a marker of unligated fragments; the authors are leaning heavily on the reader already knowing this relationship. That same sentence strongly implies that fast forks uncouple polymerases, but ref 134 shows that experimental uncoupling accelerates forks, not necessarily that fast forks uncouple the polymerases. ref 91 is cited on line 400 as showing MCM depletion causes cancer, although that reference is from a zebrafish study

and reports cell cycle defects, but not cancer. Ref 40 doesn't mention cyclin E, Ref 160 doesn't mention CDT1, etc.

Minor: Line 309 please clarify "with opposite polarity 5'-3' complementing CMG activity".

POINT-BY-POINT RESPONSE TO THE REVIEWER'S COMMENTS

We would like to thank both Reviewers for their enthusiasm, valuable comments, and excellent guidance on how to improve our Perspective. Their suggestions were very insightful, and we have diligently incorporated all their concerns in our revised manuscript. We have provided a detailed point-by-point response to each comment, outlining all the new additions and amendments that we have made.

Reviewer #1 (Remarks to the Author):

The MCM2-7 complex plays a central role in eukaryotic DNA replication. This complex binds genomic DNA during the late M to G1 phase, forming an MCM-double hexamer, the process known as licensing. It is subsequently converted into the CMG helicase for forming the replisome during the S phase. Multiple mechanisms control MCM2-7, ensuring that DNA replication takes place only once in the cell cycle to maintain genomic integrity. This review article by Yadav and Polasek-Sedlackova highlights the importance of regulating MCM2-7, with a particular focus on its high abundance compared to other replication factors. This review article is based on their previous reports (Ref. 40, Sedlackova et al. Nature, 2020; Ref. 75, Polasek-Sedlackova, Nat. Commun., 2022), as well as numerous studies done by other researchers.

The review article is well-written and provides many recent updates on MCM biology. While there are no major issues, there are some points that the authors should consider for correction, clarification or additional information.

We sincerely thank the Reviewer for appreciating our work.

1. Yeast studies have driven the mechanistic study of DNA replication. Although the authors mainly discuss DNA replication and MCM biology in mammalian cells, there are instances where findings in yeast and mammalian cells are not clearly distinguished. Here are some examples.

We agree with the Reviewer that studies using *S. cerevisiae* as a model organism represent a leading source of our knowledge regarding eukaryotic DNA replication and its regulation at the molecular level. We went carefully through the entire manuscript to make sure that we clearly distinguished between yeast and mammalian cells.

- Line 49: "MCM double hexamers, which are kept in the inactive state throughout the licensing period (Ref. 10-12)". The references are from yeast studies. It should be noted that the human MCM-double hexamer was recently reported (Li et al. Cell, 2023).

We thank the Reviewer for reminding us of an important reference, which we incorporated in the revised manuscript and highlighted in our reference list. In the introductory section, we have included relevant literature on replication licensing in human cells.

- Lines 53-54: "most recently identified DONSON as a specific firing factor in vertebrates". Ref. 24 reported DONSON in *C. elegans*. It seems that DONSON is conserved even in insects and plants.

We have now corrected this inappropriate wording in the revised manuscript.

- Lines 59-60: "The recent development of in vitro reconstitution assays of eukaryotic DNA

replication". As far as I understand, in vitro reconstitution has been performed with yeast proteins, not yet with human proteins.

We agree with the Reviewer that in vitro reconstitution assays have been performed so far with yeast proteins. This sentence was part of the introductory part summarizing key findings of eukaryotic DNA replication in general. However, to avoid any potential confusion for the reader, we decided to remove those former lines (59-60) from the introduction. Motivated by the Reviewer's comment, we included an additional forward-looking question, 'How are replication origins recognized and regulated in higher eukaryotes?' in the last paragraph, as it is conceivable that forthcoming studies could elucidate novel pathways and regulators that are exclusive to higher eukaryotes.

2. The authors describe that MCMBP chaperones the MCM hexamer formation. However, there are discrepancies with other studies regarding the assembly of MCM2-7. The authors should address these discrepancies and provide a balanced view of the findings.

"One such finding involves the identification of MCM binding protein (MCMBP) that strongly interacts with the MCM3-7 subcomplex and acts as a chaperone for nascent MCMs (Ref. 40, 60-62).

Specifically, MCMBP plays an important role in the assembly and nuclear transportation of nascent MCM3-7 subcomplexes, while the MCM2 subunit enters the nucleus autonomously (lines 126-130)".

"Additionally, it remains unclear why the MCM2 subunit is separated from the rest of the MCM subcomplex, which is chaperoned by MCMBP and transported to the nucleus (lines 134-136)".

Additionally, Figure 2b gives the impression that the assembly of MCM2-7 occurs only in the nucleus.

I understand that these are based on the findings in their publication (Ref. 40. Sedlackova et al.

Nature, 2020). However, there is no direct biochemical evidence showing that there are an MCM3-7 subcomplex and MCM2 monomer in normal conditions in this paper. There is another report showing there are subcomplexes of MCM2/4/6/7 and MCM3/5, which eventually form MCM2-7 (Ref.65, Prokhorova and Blow, JBC, 2000). In line with this study, a recent report showed that MCMBP directly binds MCM3, makes an MCM3/MCMBP subcomplex, and works to incorporate MCM3/5 into MCM2/4/6/7 (Ref. 60, Saito et al. eLife, 2022). This paper also argued that the nuclear transport function of MCMBP is not essential, and the assembly might occur both in the cytoplasm and nucleus.

We thank the Reviewer for this helpful comment. We have now included and discussed both proposed models on MCMBP function in MCM biogenesis primarily based on *Sedlackova et al., Nature, 2020* and *Saito et al., eLife, 2022* in the text (lines 129-150) as well as in graphical representation (Figure 2b). We believe that we now provide a more balanced view of the mechanism of MCM biogenesis.

Briefly, the model proposed by Saito et al. is based on the observation that MCMBP directly interacts with the MCM3 subunit and shields it from proteasomal degradation. Furthermore, the authors suggest that MCMBP may facilitate the incorporation of the MCM3/5 subcomplex into the MCM2/6/4/7 subcomplex. Although the proposed model is certainly elegant in its simplicity, it appears to deviate from a previous study conducted by *Nishiyama et al., Genes Dev, 2011*. In this study, the authors identified a direct interaction between MCMBP and the MCM7 subunit rather than MCM3 as proposed in the current model. This inconsistency may, however, suggest an additional layer of complexity in the MCMBP function during MCM ring assembly.

The second model proposed by Sedlackova et al. is based on the experiments revealing that shortly after MCMBP depletion or mutating its NLS (nuclear localization signal), nascent MCM3-7 subunits are rapidly mislocalized in the cytoplasm while MCM2 remains orphaned in the nucleus.

Different subcellular localizations of MCM subunits lead us to the assumption that MCM3-7 and MCM2 are transported across the nuclear membrane as two different units, similar to what was also reported for ORC sub-complexes (Ghosh *et al.* *J Biol Chem*, 2011). Moreover, the strong association of MCMBP with MCM3-7 subunits (in contrast to the weak association with MCM2 subunit) is aligned with previous studies (Sakwe *et al.*, *Mol Cell Biol*, 2007; Nishiyama *et al.*, *Genes Dev*, 2011). Based on this, we draw a model in which MCMBP is important for assembling nascent MCM3-7 subcomplex in the cytoplasm and contributes to its transport to the nucleus. The MCM2 subunit enters the nucleus autonomously, where the entire MCM ring is assembled. Whether separating the MCM2 subunit from the rest of the MCM subcomplex during its transport is a steric requirement or has a functional significance in regulating MCM ring assembly remains an interesting avenue for future investigations.

Regarding the nuclear transport function of MCMBP, in Sedlackova *et al.*, *Nature*, 2020, we showed that mutation of the MCMBP's NLS, in contrast to complete MCMBP depletion protects MCM3-7 subcomplexes from degradation, but their transport to the nucleus is less efficient (MCMBP-MCM3-7 relies only on NLS located on MCM3 subunit, this can also explain why MCMBP lacking NLS can be found in the nucleus). Importantly, U2OS cells expressing MCMBP with non-functional NLS showed limited origin licensing and higher levels of DNA damage. The impaired efficiency in MCM transport could primarily affect cells that need to license MCM excess at replication origins in a very short time. However, whether the MCMBP transport function is essential only in specific cells needs to be explored in the future.

Despite the gaps and discrepancies in current MCM biogenesis models, both studies came to the same conclusion that MCMBP acts as a dedicated chaperone for nascent MCMs and is important to maintain their proper equilibrium in the cellular environment. We strongly believe that the proposed models and ideas will inspire future work to elucidate the molecular mechanism of MCM biogenesis further.

3. In the section on MCM excess as an integral component of replication fork speed control, the authors give the impression that fork speed is mainly controlled by obstacles ahead of the forks. However, it is well known that fork speed is strongly affected by the number of active origins and dNTP pools (Anglana *et al.* *Cell*, 2003; Poli *et al.* *EMBO J.* 2012; Zhong *et al.* *JCB*, 2013; Rodriguez-Acebes *et al.* *JBC*, 2018). The authors should acknowledge this point.

We have highlighted this in the revised version of the manuscript (lines 326-328).

4. In the last section mentioning cancer development and therapy, there is a famous genetic disorder called Meier-Gorlin syndrome (MGORS) that the causative genes are found in many replication proteins, including MCM5 and 6. This point is relevant and should be included.

We have now included Meier-Gorlin disease in the revised manuscript (lines 400-406). However, to maintain the primary focus of the paragraph on how deregulated origin licensing contributes to oncogenic transformation, we decided to refer to a recent excellent review written by Schmidt and Bielinsky, who describe in great detail clinical phenotypes and functional mechanisms of congenital diseases associated with dysfunctional DNA replication.

Reviewer #2 (Remarks to the Author):

In this Perspective, Yadav & Polasek-Sedlackova discuss how adequate origin licensing is a crucial step to ensure faithful DNA replication in S phase. They summarize the fundamental studies that

established in the field mechanisms of origin licensing and firing, as well as some recent studies on MCM equilibrium and factors affecting origin selection in G1, and origin firing in S phase. They also discuss in detail dormant origins, which is an important safeguard mechanism against genome instability. The figures and table are attractive and helpful. This Perspective is very suitable for publishing in *Communications Biology*, and will be of interest to readers who are particularly interested in understanding more about origin licensing and firing, DNA replication, and genome stability. The authors are to be commended on a thorough presentation!

We sincerely thank the Reviewer for his/her appreciation for our work.

We noted several places where word-choice or phrasing could lead to confusion. The authors can consider making changes for clarity:

1. The protein concentrations in the Table are very useful for the field to have collected. However, line 151 refers to that paper as using single cell proteomics; the technology is not yet available to analyze whole proteomes of single (human) cells.

We apologize for the inappropriate wording. We have now corrected it in the revised manuscript.

2. MCM complexes are referred to as helicases throughout the article – even in G1 phase -, although they are not yet helicases until they become part of the active CMG enzyme in S phase.

We have corrected the wording in the revised manuscript to ensure that the term helicase is used only when referring to active CMG enzyme.

3. Line 142 – please clarify what is meant by the “soluble fraction”- as written it implies interactions away from DNA altogether, but the authors may have meant histones in their (bound) transit from parental to daughter strands.

Based on the suggestion of Reviewer #1 (please see our response to his/her comment nr. 2), we have decided to provide a more balanced view of the mechanism of MCM biogenesis by discussing in great detail novel models proposed by *Saito et al., eLife, 2022* and *Sedlackova et al., Nature, 2020* (lines 129-150 in the main text). In the new version, we avoid this inappropriate wording.

4. The percentage of MCM that is loaded is influenced by MCM expression, and some tumor-derived cell lines express enough MCM that a substantial fraction remains unloaded. The authors may consider the consequences of different origin licensing protein expression levels in different cell types for this concept that most of the MCM is loaded. It may be worth mentioning that the HeLa cells used for the study supporting Table 1 have deregulated expression of most replication proteins.

We thank the Reviewer for this important comment. We have now included two sentences acknowledging these points in the main text, lines 95-96, and in the Table description.

5. The term “MCM excess” is used frequently, but the authors mean an excess of loaded MCM, and not necessarily just more MCM expression; they could avoid that potential confusion with minor wording changes.

We thank the Reviewer for the helpful comment. We have changed the wording to avoid potential confusion.

6. In panel 2C, the figure labeling indicates that the reduction in total MCM chromatin loading is from the removal of excess MCMs. During S phase the active MCMs are also removed as forks terminate, so this label could be misleading.

We apologize for the confusion; we have now corrected Figure 2c, indicating the release of both inactive and active MCMs.

Finally, The authors are encouraged to clarify how some citations directly match facts of the sentence in which they occur. For example, Line 333 refers to studies demonstrating fast fork progression, but ref 131 doesn't measure Okazaki fragments, only PARP dependence which is a marker of unligated fragments; the authors are leaning heavily on the reader already knowing this relationship. That same sentence strongly implies that fast forks uncouple polymerases, but ref 134 shows that experimental uncoupling accelerates forks, not necessarily that fast forks uncouple the polymerases. ref 91 is cited on line 400 as showing MCM depletion causes cancer, although that reference is from a zebrafish study and reports cell cycle defects, but not cancer. Ref 40 doesn't mention cyclin E, Ref 160 doesn't mention CDT1, etc.

We apologize for any confusion caused by the inaccuracies in the references. These inaccuracies arose very likely due to slight modifications made in the main text during the final editing process. However, we have addressed this issue by eliminating all inaccuracies from the revised manuscript and ensured that all references are properly cited throughout.

Minor: Line 309 please clarify "with opposite polarity 5'-3' complementing CMG activity".

We have clarified our statement in the revised manuscript.

REVIEWERS' COMMENTS:

Reviewer #1 (Remarks to the Author):

The authors have done a great job. The MS is well-balanced with lots of updates. Many congratulations!

Reviewer #2 (Remarks to the Author):

We are now satisfied with the revised version and congratulate the author on a valuable contribution to the field.

POINT-BY-POINT RESPONSE TO THE REVIEWER'S COMMENTS

We would like to thank both reviewers for their enthusiasm, valuable feedback, and excellent guidance during the revision process. We are pleased to report that both reviewers are satisfied with our revision and do not have additional specific comments.

Reviewer #1 (Remarks to the Author):

The authors have done a great job. The MS is well-balanced with lots of updates. Many congratulations!

Reviewer #2 (Remarks to the Author):

We are now satisfied with the revised version and congratulate the author on a valuable contribution to the field.